# Exploring a Cost-Efficient Model for Predicting Cerebral Aβ Burden Using MRI and Neuropsychological Markers in the ADNI-2 Cohort

**DOI:** 10.3390/jpm10040197

**Published:** 2020-10-27

**Authors:** Hyunwoong Ko, Seho Park, Seyul Kwak, Jungjoon Ihm

**Affiliations:** 1Interdisciplinary Program in Cognitive Science, Seoul National University, Seoul 100-011, Korea; powerzines@snu.ac.kr; 2Dental Research Institute, School of Dentistry, Seoul National University, Seoul 100-011, Korea; 3Department of Psychiatry, Seoul National University College of Medicine & SMG-SNU Boramae Medical Center, Seoul 156-707, Korea; 4Behavioral Neuroscience Program, School of Medicine, Boston University, Boston, MA 02101, USA; sehopark@bu.edu; 5Office of Dental Education, School of Dentistry, Seoul National University, Seoul 100-011, Korea

**Keywords:** amyloid beta, neuropsychological assessment, machine learning, Alzheimer’s disease

## Abstract

Many studies have focused on the early detection of Alzheimer’s disease (AD). Cerebral amyloid beta (Aβ) is a hallmark of AD and can be observed in vivo via positron emission tomography imaging using an amyloid tracer or cerebrospinal fluid assessment. However, these methods are expensive. The current study aimed to identify and compare the ability of magnetic resonance imaging (MRI) markers and neuropsychological markers to predict cerebral Aβ status in an AD cohort using machine learning (ML) approaches. The prediction ability of candidate markers for cerebral Aβ status was examined by analyzing 724 participants from the ADNI-2 cohort. Demographic variables, structural MRI markers, and neuropsychological test scores were used as input in several ML algorithms to predict cerebral Aβ positivity. Out of five combinations of candidate markers, neuropsychological markers with demographics showed the most cost-efficient result. The selected model could distinguish abnormal levels of Aβ with a prediction ability of 0.85, which is the same as that for MRI-based models. In this study, we identified the prediction ability of MRI markers using ML approaches and showed that the neuropsychological model with demographics can predict Aβ positivity, suggesting a more cost-efficient method for detecting cerebral Aβ status compared to MRI markers.

## 1. Introduction

In 2010, 35.6 million people across the world were estimated to have dementia, which will approximately double to 65.7 million in 2030 [1]. With growing numbers of patients with dementia, the cost of dementia will become more problematic. In the US, the annual payment for dementia care was USD 236 billion in 2016, being expected to increase to more than USD 1 trillion in 2050 [2]. Alzheimer’s disease (AD) is the most common cause of dementia, explaining about 70% of dementia cases [3]. Structural magnetic resonance imaging (MRI) has been used most frequently as a diagnostic modality for AD. It can show structural pathological changes in the brain and irreversible atrophy due to the neurodegeneration of AD [4]. A decade ago, it was thought that people with dementia had AD-related pathological changes and those who were cognitively normal (CN) did not have such changes in their brains [5]. However, this has been reconsidered, and now, it is thought that both AD-related pathological changes and clinical symptoms depend on the stage of disease, suggesting that the end state of the process is dementia, with the accumulation of pathological and clinical changes. In this respect, structural MRI (e.g., cortical atrophy) can no longer provide specific evidence for AD, but the extent of cortical atrophy measured by structural MRI still associates closely with Braak staging at autopsy [6,7].

Recently, it has been well established that beta-amyloid (Aβ) is an essential component of AD, and Aβ begins to accrue almost 10–15 years before the clinical onset of AD [8]. Advances in AD research and imaging methods have made it possible to detect cerebral Aβ deposition in people without postmortem autopsy by using positron emission tomography (PET). Approximately 20–30% of CN elderly have AD-specific neuropathology in their brains as measured by carbon-11-labeled Pittsburgh compound-B (^11^C-PiB) PET [9]. However, these techniques have a few limitations. Employing amyloid PET is expensive, it is not commonly available, and it causes exposure to radiation. Cerebrospinal fluid (CSF) analysis can also be used to gauge Aβ 42; however, it is labor intensive, requires an invasive lumbar puncture, and has poor inter-laboratory reliability, making it difficult to implement [10].

Neuropsychological assessment, a sensitive and cost-efficient measurement for diagnosing AD, can be used as a non-invasive evaluation tool for AD-related pathology. Despite several advantages of using neuropsychological assessment for evaluating AD pathology, only a few studies have used the assessment as a predictive tool [11,12]. A recent study demonstrated the ability to predict based on neuropsychological assessment and demographics using a brief machine learning (ML) approach [13]. In the study, the ML result suggested that neuropsychological tests with demographics could be used as a preliminary screening for detecting Aβ before neuroimaging, such as amyloid PET. Except for several studies, the majority of the studies in the AD-related research fields that applied ML used structural MRI images [14,15]. Additionally, there were few studies that focused on applying ML techniques to AD studies to predict Aβ status [16,17].

The goal of the current study was to propose models that can predict an abnormal level of cerebral Aβ based on neuropsychological test scores and demographics using several ML algorithms. For exploring and comparing the prediction performance of neuropsychological test-derived models, two brain measures that are well known to be related to cerebral amyloid deposition were considered [18,19]: cortical thickness and volume. Finally, feature-selection approaches were used to provide the explainable and practical implications of the results.

## 2. Materials and Methods

### 2.1. Ethics Statement

In this study, we used participant data from the Alzheimer’s Disease Neuroimaging Initiative (ADNI), a multicenter project with approximately 50 medical centers and university sites across the United States and Canada [20]. The ADNI was launched in 2003 as a public–private partnership led by principal investigator Michael W. Weiner, MD. Its primary goal is to test whether biological markers such as serial MRI, PET, and clinical and neuropsychological assessment can be combined to measure the progression of patients with mild cognitive impairment (MCI) and early AD. The participants were between 55 and 90 years old, and were able to undergo all the assessment procedures and consent to participate in longitudinal follow-up. Written consent forms were obtained from all participants, and the study was conducted after approval from the Institutional Review Board at each participating institution.

### 2.2. Participants

CN participants were the control group in the ADNI study and showed no significant clinical symptoms such as depression, MCI, or dementia. The participants had a significant memory concern (SMC) score within the normal range for cognitive function but reported concerns about their memory. Participants with early MCI (EMCI) and late MCI (LMCI) reported their SMC either autonomously via an informant or through clinicians. However, daily living activities were preserved, and other cognitive domains showed no significant impairment, without signs of dementia. The degree of MCI (early or late) was determined with the Logical Memory subtest of the Wechsler Memory Scale [21]. Participants with AD met the National Institute of Neurological and Communicative Disorders and Stroke–Alzheimer’s Disease and Related Disorders Association criteria for probable AD [22,23]. A detailed description of the inclusion/exclusion criteria can be found at ADNI website [24].

Data were downloaded from the ADNI database and included all the subjects recruited in the ADNI-2 with complete baseline data available for cognitive assessment, Apolipoprotein E (APOE) genotype processing, and PET Aβ quantitation. Our study sample included 724 subjects (170 control subjects, 95 with SMC, 324 with MCI (169 with EMCI and 155 with LMCI), and 135 with AD) who were recruited between 2011 and 2013, each of whom had a baseline APOE genotype, structural MRI, and florbetapir session.

### 2.3. Amyloid PET Data

Baseline Aβ deposition was visualized using florbetapir-PET. Amyloid PET results were downloaded from the latest available dataset “UCSF Berkeley—AV45 Analysis” (accessed on 27 August 2019) in the ADNI database. Detailed methods for PET acquisition and analysis can be found in the work by Landau and colleagues [25,26]. The florbetapir images consisted of 4 × 5 min frames acquired 50–70 min after injection. The images were realigned, averaged, resliced to a common voxel size (1.5 mm), and smoothed to a common resolution of 8 mm in full width at half-maximum [27]. Structural T1-weighted images were acquired concurrently with the baseline florbetapir images. The images were used as structural templates to define the cortical regions of interest (ROIs), and the reference regions in native space for each subject were obtained with the FreeSurfer (version 5.3.0, Boston, MA, USA), which is documented and freely available online [28]. Baseline florbetapir scans for each participant were co-registered to baseline structural T1-weighted images. The images were subsequently used to extract weighted cortical retention indices and the standardized uptake value (SUV) from gray matter within four cortical ROIs (frontal, anterior/posterior cingulate, lateral parietal, and lateral temporal) that were averaged to generate a mean cortical SUV as described online in greater detail [29]. Cortical SUV ratios (SUVRs) were obtained by normalizing the cortical SUV to the mean uptake in the whole cerebellum reference region. The participants were classified as cerebral Aβ positive if the florbetapir SUVR was greater than 1.1.

### 2.4. Structural MRI Data

Structural MRI scans were obtained at baseline visits according to a standardized protocol [30]. Briefly, all the participants in the current study were scanned using the 3T-MRI scanning protocol (GE Healthcare, Milwaukee, WI, USA; Philips, Eindhoven, Netherlands; Siemens, Erlangen, Germany). Cortical reconstruction and volumetric segmentation were performed with the same FreeSurfer image analysis suite. Cortical thickness and volume data were downloaded from “UCSF—Cross-Sectional FreeSurfer (5.1) (accessed on 8 November 2019)” in the ADNI database. Further details for MRI scans are described in more detail elsewhere [31].

### 2.5. Neuropsychological Assessment

The following measurements were considered: the Mini-Mental State Examination (MMSE) [32], Alzheimer’s Disease Assessment Scale (ADAS-Cog 13) [33], the Rey Auditory Verbal Learning Test (AVLT) [34], Logical Memory (LM) [21], the Clock Drawing Task [35], the Trail Making Test [36], Category Fluency [37], the Boston Naming Test (BNT) [35], and the American National Adults Reading Test (ANART) [38].

### 2.6. Statistical Analyses

The demographics were compared between groups using t-tests. χ^2^ tests were used to compare dichotomous variables. For predictability estimation, the aforementioned features, cortical thickness and volume, and cognitive performance were used as input data for the supervised learning ML algorithm. The algorithms used in current study were as follows: logistic regression (LR), support vector machine (SVM), boosted decision tree (BDT), and artificial neural network (ANN). Since the over-fitting problem at the second level of inference (i.e., model selection) could cause a deleterious effect on generalizability, model selection for the ML models was not conducted in the current study [39].

Next, to explain multivariate profiles with respect to input features that were accurately distinguished from Aβ positivity for participants, the adaptive least absolute shrinkage and selection operator (LASSO) ML algorithm were applied to the dataset [40]. The adaptive LASSO, which is a penalized regression method [41], is a popular technique for simultaneous estimation and consistent variable selection [40]. The regression coefficients of unimportant variables shrank to 0 upon implementing the adaptive LASSO. In that regard, the adaptive LASSO algorithm provided interpretable results related to abnormal levels of cerebral Aβ.

In order to evaluate the generalizability of the results from the ML algorithms, 10-fold cross-validation was applied during the variable selection process. First, the data were randomly split into a training set (66.7% of the data) and a test set (33.4% of the data). All of the ML models were fitted using the training set, and classifications were separately made on the test and training datasets. The optimal parameter, lambda, was determined across 1000 iterations of 10-fold cross-validation to minimize the deviance of the model. Then, predictions were made on the test set based on the ML models trained in the training set.

The parameters used to compare the prediction performance of and validate the models were as follows: the accuracy (ACC), precision (PRE), recall (REC), F1 score, and area under the curve (AUC). Analyses were performed using R software (version 3.6.1, Vienna, Austria) [42].

## 3. Results

### 3.1. Subject Characteristics

The demographic data for all participants are presented in Table 1. Of the 724 study participants, 339 (46%) were APOE ε4 carriers and 400 (54%) were Aβ positive (Aβ+).

### 3.2. Feature Combination and Performance of the ML Models

In an attempt to compare the predictive performance of each model across the combinations of input features, we split the features into several subsets: (1) whole dataset: demographics (Demo) + neuropsychological assessment (NA) + cortical thickness (CT) + cortical column (CV); (2) Demo + CV + NA; (3) Demo + CT + NA; (4) Demo + CV + CT; (5) Demo + NA. For the overall values, it was difficult to show that there was a significant difference in the prediction ability of each performance index or applied model among the combinations using MRI markers (Table 2). In this respect, we therefore verified the pattern of the prediction ability without MRI markers, i.e., the fifth combination. Table 3 shows the results using the fifth combination. The combination with 16 neuropsychological assessment features, excluding MRI markers, produced results similar to those with up to 270 MRI markers. Thus, only the fifth combination was considered in the subsequent analysis for feature selection.

### 3.3. Feature Selection from Adaptive LASSO

The demographic (age, gender, and APOE ε4 status) and neuropsychological data were assessed for their ability to predict cerebral amyloid positivity. The adaptive LASSO, based on the shrinkage method, was conducted to identify significant features among the 19 variables for predicting Aβ positivity. Figure 1 shows the multivariate profiles for cerebral Aβ positivity, revealed by the adaptive LASSO algorithm. First, Aβ positivity was more prevalent in participants who were older, female, and APOE ε4 carriers and showed poor cognitive performance in several measures in the ADNI-2 population (worse delayed recall and global cognition: ADAS-Cog 13). The receiver operating characteristic (ROC) demonstrated an AUC of 0.824 for distinguishing Aβ positivity using 6 of 20 features (Figure 2).

## 4. Discussion

The present study proposed an ML approach using various classifiers based on demographics, cortical thickness and volume, and neuropsychological performance data to predict cerebral Aβ deposition in the ADNI-2 cohort. The subset using demographics and neuropsychological assessment data demonstrated better cost-efficiency compared to other models including structural MRI data. Therefore, the cost-efficient model was chosen for further analysis, and the adaptive LASSO algorithm was adopted in the feature-selection process, which was the most predictive measure for cerebral Aβ positivity. Finally, multivariate profiles of cognitive performance and demographics were used to support the explainability of the model.

Although some studies have reported high predictive accuracy for Aβ positivity based on ML methods, these studies were based on blood biomarkers or combined neuropsychological tests [16,17]. Only one recent study investigated the predictability of Aβ status using neuropsychological assessment based on ML [13]. To our knowledge, this is one of the first studies to evaluate relative performance using ML models predicting cerebral amyloid status based on several ML algorithms in AD research.

Current findings demonstrate that including structural MRI markers can predict Aβ positivity with an 85% prediction ability (Demo + CV + NA model using SVM, Table 2) in the ADNI-2 participants. The cost-efficient model (Demo+NA), however, shows similar results (Table 3), leading to the same result using the adaptive LASSO model (AUC = 0.85). Given these results, specific neuropsychological measures, even without neuroimaging markers, have practical implications for the prediction of neuropathological biomarkers in AD without invasive methods (i.e., PET or CSF methods).

In the ADNI-2 cohort, age, gender, and APOE ε4 status are significant variables for predicting Aβ positivity. However, education is not a significant indicator. These results are consistent with previous studies [43,44,45]. It is remarkable that certain cognitive measures show a higher coefficient for predicting Aβ positivity compared to that of age, which is consistent with previous studies [13,46].

As mentioned earlier, most of the studies using ML approaches have focused on the diagnosis and progression of the disease based on limited biomarkers such as volumetric and/or surface-based brain measures, or blood proteins in AD research [14,15,47]. In that regard, our findings show that ML-based models can predict Aβ positivity, suggesting a potential application of Aβ assessment through non-invasive MRI markers. Since the National Institute on Aging and Alzheimer’s Association (NIA-AA) research framework has been updated, neuroimaging markers derived from structural MRI scans have been considered as non-specific pieces of information about neuronal injury and neurodegenerative changes in the AD course [48]. However, our results show that MRI markers may still provide information for AD-specific pathology with a combination of other layers of information, such as demographics and neuropsychological assessments. Given the fact that amyloid PET scans have relatively limited accessibility compared to MRI scans, predicting cerebral Aβ status using MRI scans may be more feasible in the health care system. First, the progress of AD pathology can be predicted during the assessment of other brain diseases with an automated ML algorithm for online structural MRI analysis. Second, it could be an alternative modality for assessing AD-specific pathology if the use of amyloid PET scans is limited.

More importantly, our findings demonstrate that results based on neuropsychological data with demographic data are nearly as accurate as the MRI markers in predicting cerebral Aβ status. This suggests the following benefits. Given that in most AD-related clinical trials, delayed recall and the global cognition assessment are administrated as screening tools for participants’ inclusion, a combination of these tests can be a preliminary tool for predicting Aβ status. For ADAS-Cog, previous studies showed that a poor performance on the test suggested Aβ-related cognitive decline [49] or was suitable for inclusion criteria for clinical trials for the population of those who had Aβ1-42 pathology [50]. This suggests that ADAS-Cog needs to be considered in clinical trials or clinics in order to detect AD pathology, as well as using the MMSE, which is commonly used to screen for dementia. A deficit in episodic memory, such as poor performance on the AVLT, has previously been widely used to verify cognitive impairment in AD [51]. In addition, a recent study found that females with high Aβ levels showed faster decline in the verbal list learning test, suggesting that the study supports our results of an adaptive LASSO multivariate profile [52]. Another episodic memory test, but in a different form, is LM, which is known as a neuropsychological marker for capturing early changes in AD and is also used as a marker for longitudinal cognitive change [53,54]. Overall, these findings have implications for which tests are useful in finding AD-specific signatures, given the neuropsychological battery. In addition, there are financial advantages of detecting cerebral Aβ without amyloid PET scans, considering the high cost of these scans (USD 5000 per person in the US [55]). Moreover, the explainable profile from the adaptive LASSO results can provide clinicians with reliable information about AD-specific pathology.

The current study has several limitations. First, the ADNI population is not representative of the general population because the participants were deliberately selected based on robust inclusion criteria. Moreover, most participants in the ADNI-2 cohort were recruited only from North America. Second, the results were only estimated from a baseline visit; hence, the cross-sectional results are limited for causal assumptions of the input features and the outcome variables. In this regard, 10-fold-cross validation for out-of-sample generalization may support the generalization of the results. Third, since the ADNI participants were recruited according to a robust clinical design for the AD spectrum, applying the implications to general clinics should be conducted with caution. Fourth, the use of limited structural MRI markers may not have enriched the results, suggesting that the consideration of other MRI markers such as resting state functional MRI, hippocampal volume, or T2-weighted image markers needs to be considered for further analysis. Finally, although the results may cause an over-fitting problem regarding the high dimensionality in structural MRI data, the process of dimension reduction was not implemented. Dimension reduction was not performed in order to prevent the loss of information because it was pre-processed by the FreeSurfer suite rather than 2D image data. Undoubtedly, since the feature-selection procedure for the final model serves as a dimensionality reduction, it was assumed that the cost-efficient model could have avoided over-fitting problems. Further research should be validated against various cohort studies and longitudinally designed studies with various other clinical features of AD.

## 5. Conclusions

In conclusion, our findings demonstrate that MRI markers using an ML algorithm might predict an abnormal level of Aβ in an ADNI-2 cohort without invasive methods. The results also show the same prediction ability using a neuropsychological model based on the adaptive LASSO methods, which is cost-efficient compared to conventional neuroimaging methods. The application of these findings may provide us with the opportunity to develop a precise intervention for patients who suffer from dementia. The cost-efficient, non-invasive, and easily accessible aspects of the method could contribute to thorough recruitment for AD clinical trials and serve as an alternative method for patients who have difficulties in undergoing PET scans.

## Figures and Tables

**Figure 1 jpm-10-00197-f001:**
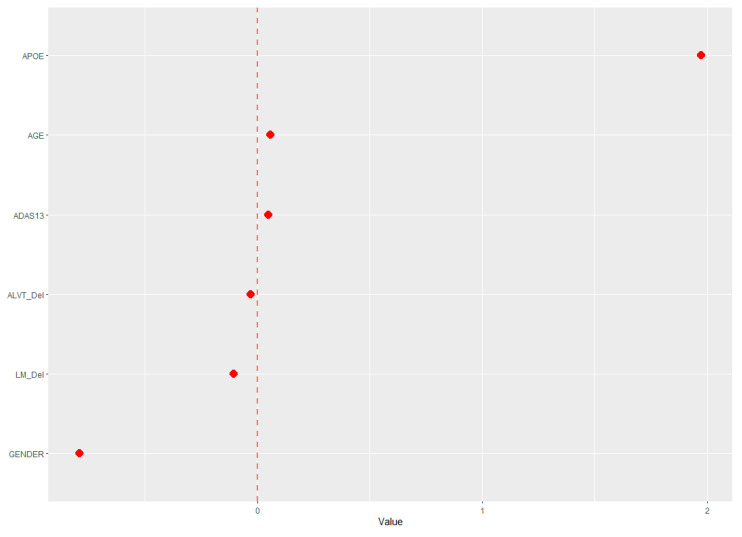
Multivariate patterns of demographic information and neuropsychological markers predicting cerebral Aβ burden: ADAS13, Alzheimer’s disease assessment scale; APOE, ApoE ε4 positivity; AVLT _Del, Rey auditory verbal learning test delayed recall; LM_del, logical memory delayed recall.

**Figure 2 jpm-10-00197-f002:**
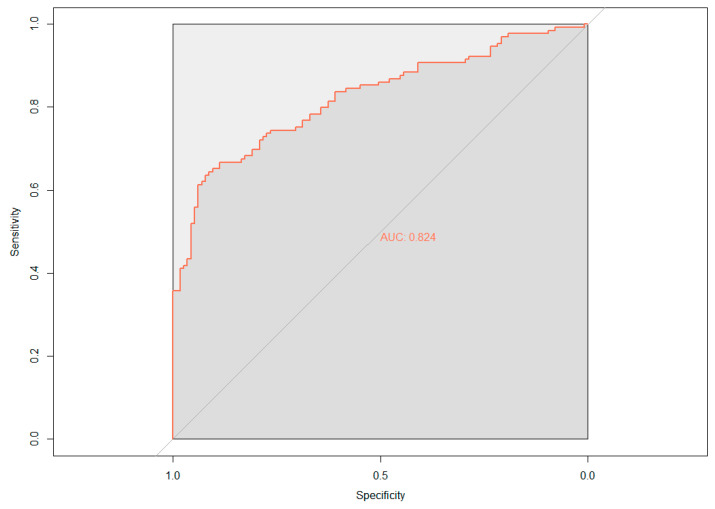
Receiver operating characteristic (ROC) curve based on the adaptive least absolute shrinkage and selection operator (LASSO) result; AUC, area under the curve.

**Table 1 jpm-10-00197-t001:** Demographics of participants.

Characteristics	CN (*N* = 170)	SMC (*N* = 95)	EMCI (*N* = 169)	LMCI (*N* = 155)	AD (*N* = 135)	Total (*N* = 724)
Age, years	73.9 (6.2)	72.9 (5.7)	71.6 (7.1)	72.6 (7.6)	74.9 (8.1)	73.2 (7.1)
No. of females (%)	91 (51%)	55 (58%)	71 (42%)	71 (46%)	56 (41%)	344 (47%)
Education, years	16.6 (2.5)	16.7 (2.6)	16.2 (2.7)	16.6 (2.6)	15.7 (2.6)	15.7 (2.6)
No. of APOE ε4 carriers (%) ^a^	50 (28%)	29 (31%)	77 (46%)	90 (58%)	93 (69%)	339 (46%)
Aβ positivity (%)	53 (29%)	33 (35%)	84 (50%)	108 (70%)	122 (90%)	400 (54%)

Abbreviations: APOE—Apolipoprotein; Aβ—Amyloid beta; CN—Clinically normal; SMC—Subjective memory concerns; EMCI—Early mild cognitive impairment; LMCI—Late mild cognitive impairment; AD—Alzheimer’s disease. Data are presented as mean (SD) unless otherwise indicated. ^a^ APOE ε4 carriers are the percentage of individuals with at least one APOE ε4 allele.

**Table 2 jpm-10-00197-t002:** Results of models including MRI markers.

	ACC	PRE	REC	F1	AUC
**WHOLE**					
LR	0.75	0.78	0.76	0.77	0.84
SVM	0.75	0.79	0.75	0.77	0.85
BDT	0.76	0.81	0.72	0.77	0.85
ANN	0.74	0.77	0.74	0.76	0.81
**Demo + CV + NA**					
LR	0.76	0.80	0.76	0.78	0.84
SVM	0.76	0.80	0.74	0.77	0.85
BDT	0.75	0.81	0.71	0.76	0.84
ANN	0.73	0.77	0.73	0.75	0.81
**Demo + CT + NA**					
LR	0.75	0.79	0.75	0.77	0.84
SVM	0.76	0.80	0.76	0.78	0.84
BDT	0.77	0.83	0.74	0.78	0.82
ANN	0.75	0.77	0.78	0.78	0.82
**Demo + CV + CT**					
LR	0.77	0.82	0.76	0.79	0.84
SVM	0.74	0.80	0.72	0.76	0.84
BDT	0.75	0.83	0.70	0.76	0.83
ANN	0.71	0.85	0.58	0.69	0.81

Abbreviations: WHOLE—Demp + NA + CT + CV; Demo—Demographics; NA—Neuropsychological assessment; CT—Cortical thickness; CV—Cortical volume; ACC—Accuracy; PRE—Precision; REC—Recall; AUC—Area under curve; LR—Logistic regression; SVM—Support vector machine; BDT—Boosted decision tree; ANN—Artificial neural network.

**Table 3 jpm-10-00197-t003:** Results of models excluding MRI markers.

	ACC	PRE	REC	F1	AUC
**Deom + NA**					
LR	0.75	0.79	0.74	0.77	0.83
SVM	0.75	0.80	0.73	0.76	0.83
BDT	0.71	0.76	0.70	0.73	0.79
ANN	0.75	0.83	0.70	0.75	0.83

Abbreviations: Demo—Demographics; NA—Neuropsychological assessment; ACC—Accuracy; PRE—Precision; REC—recall; AUC—Area under curve; LR—Logistic regression; SVM—Support vector machine; BDT—Boosted decision tree; ANN—Artificial neural network.

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
