# Peer review of "Exploring a Cost-Efficient Model for Predicting Cerebral Aβ Burden Using MRI and Neuropsychological Markers in the ADNI-2 Cohort"

_jpm, 2020, doi:10.3390/jpm10040197_

Round 1

Reviewer 1 Report

In this manuscript, the authors conducted studies on Aβ status prediction in AD patients using machine learning (ML) approaches. By analyzing demographic variables, MRI data, neuropsychological performance with multiple ML algorisms, they showed that 1) analysis of MRI markers using an ML algorithm might be able to predict abnormal level of Aβ status; 2) LASSO-based neuropsychological model also works well in predicting Aβ status. These findings may provide cost-efficient and non-invasive approaches for diagnosis of AD and help develop precise intervention and clinical trials for AD patients.

Overall, the study was reasonably designed, and the results are convincing and could be very useful for patients suffering from AD. I recommend it for publication after better English editing since there are many grammatical errors throughout the manuscript.

Reviewer 2 Report

Ko et al. Show that machine learning can be used as a predictor of Aß-pathology in the ADNI-2 cohort. These results have been obtained using cross-sectional data from baseline PET, MRI and neuropsychological assessment. There are a few minor comments to consider:

1.) please check for spelling mistakes

2.) Can authors describe in more detail the results why there were a few differences in performance metrics over subsets and applied algorithms. Table 2 is the overall representation of the data. It could be more understandable for the reader to present written results from each of the values that were achieved and point out the significants ones within the text.

3.) How significant is the difference seen with MRI and without? The authors state “We can see that the results of 5th combination (Demo+NA) were not much different from the results using structural MRI information” on what is this based? Which statistical test was used for comparison? Could the authors use the same test for comparing combinations in Table 5 and is the difference significant? Or are all combinations not different from each other?

4.) It would be important to clearly state why LASSO has been chosen only for Aß pathology prediction. How does that affect comparability of the data?

5.) The authors should discuss how good this model is in terms of the cross-sectional data it is based on. Is there longitudinal data in which they could validate their results?

6.) The authors should give more details for the neuropsychological (NP) results obtained. From a clinic perspective when using NP and demographics without MRI for prediction the overall presentation of NP within this cohort is important.
